# On Quantization of a Slowly Rotating Kerr Black Hole in Teleparallel Gravity

**Sérgio Costa Ulhoa [1,2,\*] , Ednardo Paulo Spaniol [3,4] and Ronni Geraldo Gomes Amorim [5]**

1   Instituto de Física, Universidade de Brasília, Brasília, DF 70910-900, Brazil
2   International Center of Physics, Instituto de Física, Universidade de Brasília, Brasília, DF 70910-900, Brazil
3   UDF Centro Universitário, Brasília, DF 70390-045, Brazil; spaniol.ep@gmail.com
4   Faculdade de Tecnologia e Ciências Sociais Aplicadas, Centro Universitário de Brasília (UniCEUB), Brasília, DF 70790-075, Brazil
5   Faculdade Gama, Universidade de Brasília, Brasília, DF 70910-900, Brazil; ronniamorim@gmail.com
*   Correspondence: sc.ulhoa@gmail.com

**Abstract:** In this article we calculate the total angular momentum for Kerr space-time for slow rotations in the context of teleparallel gravity. In order to analyze the role of such a quantity, we apply Weyl quantization method to obtain a quantum equation for the z-component of the angular momentum density, and for the squared angular momentum density as well. We present an approximate solution using the Adomian decomposition method (AM), which reveals a discrete characteristic for angular momentum.

**Keywords:** quantum gravity; teleparallelism; angular momentum

---

## 1. Introduction

In the old quantum theory, the discrete nature of physical systems played an important role. Such a feature evolved, giving rise to the so called quantum mechanics. In this process, the passage from classical theory to its quantum counterpart was developed by quantization techniques. Perhaps one of the first of such rules was the Bohr-Sommerfeld quantization [1–5].

$$\oint p_\mu dx^\mu = n\hbar.$$

In fact, such a quantization rule was applied to the Hydrogen spectral lines to obtain an atomic model. Hence, the quantization of angular momentum in Bohr's model was fundamental to the establishment of the whole quantum mechanics. Quantum theory has reached enormous acceptance in Physics because it is in agreement with the experimental data of the microscopic world. On the other hand, general relativity is equally successful in explaining macroscopic phenomena, especially those on a large scale. These two fundamental branches of physics have not yet been unified because of mutually exclusive approximations, such that in the microscopic world the gravitational force is negligible and in the macroscopic world the quantum phenomena have no effect. Thus, the quest for a quantum theory of gravitation becomes natural and extremely desirable. From the point of view of this unification, general relativity has proved flawed, at least refractory to the applications of quantization techniques. The root of this behavior lies in the old problem of gravitational energy. Since the birth of general relativity we have sought an energy-momentum tensor associated only with the gravitational field. Although there are several proposals with interesting characteristics, there is no gravitational energy expression in the scope of the metric formulation that

brings together the invariance with respect to the coordinate transformations and the dependence with the choice of the reference system.

Teleparallel gravity is an alternative theory to general relativity that predicts the same experimental results of the former. It is a theory dynamically equivalent to general relativity which has been introduced by Einstein as an attempt to construct an unified field theory [6]. In the framework of teleparallel gravity there is a well defined expression for a gravitational energy-momentum tensor [7–9], which also defines a gravitational angular momentum [10,11]. However, such an expression is not defined in phase space, which demands a great effort to apply the canonical quantization rules. This forces one to use alternative methods such as Dirac method or Weyl's quantization [12,13]. Particularly, the Weyl's method was used to obtain a discrete spectrum of mass for Schwarzschild space-time using teleparallel energy [14]. However, a quantization of gravitational angular momentum is still lacking. In this article we want to explore such a calculation for a slowly rotating Kerr space-time.

This article is divided as follows. In Section 2 the teleparallel gravity is described, the gravitational energy-momentum is introduced as well as the gravitational angular momentum. In Section 3 we apply a quantization technique to gravitational angular momentum density to obtain an eigenvalue equation for the respective operator and its square. We then give an approximated solution using the Adomian method. Finally, in the last section we present our last comments. In this article we use natural unities unless otherwise stated.

## 2. Teleparallel Gravity

Teleparallel gravity is an alternative theory of gravitation dynamically equivalent to general relativity. It is constructed out of tetrad field rather than metric tensor. The tetrad field relates two symmetries in space-time, Lorentz transformations, and passive coordinate transformations. In order to tell them apart we use latin indices $a = (0), (i)$ to designate SO(3,1) symmetry and greek indices to diffeomorphism, $\mu = 0, i$. Thus

$$
\begin{aligned}
g^{\mu\nu} &= e^{a\mu} e_a{}^\nu \, ; \\
\eta^{ab} &= e^{a\mu} e^b{}_\mu \, ,
\end{aligned}
\tag{1}
$$

where $\eta^{ab} = diag(-+++)$ is the metric tensor of Minkowski space-time. This means that for every metric tensor there are infinity tetrads, each of them is adapted to a specific reference frame. It should be noted that the tetrad field has 16 independent components and the metric only 10. These six components are totally arbitrary and they define the kinematical state of the observer once the components $e_{(0)}{}^\mu$ are associated to the 4-velocity of the observer $U^\mu$. Such an association is possible because the components $e_{(0)}{}^\mu$ remain tangent along the trajectory in a given world-line. Teleparallel gravity is not only formulated in terms of tetrad fields but it is also defined in a Weitzenböck geometry. Let us see how the equivalence to general relativity is obtained.

A Weitzenböckian manifold is endowed with the Cartan connection [15], $\Gamma_{\mu\lambda\nu} = e^a{}_\mu \partial_\lambda e_{a\nu}$, which has a vanishing curvature tensor. This feature allows one to compare vectors at different points in space-time, hence it is possible to have parallelism at distance also known as teleparallelism. On the other hand, the torsion associated to such a connection is

$$
T^a{}_{\lambda\nu} = \partial_\lambda e^a{}_\nu - \partial_\nu e^a{}_\lambda \, .
\tag{2}
$$

The Christoffel symbols $^{0}\Gamma_{\mu\lambda\nu}$ are torsion free and exist in a Riemannian geometry, thus the curvature tensor plays all dynamical roles for metric theories of gravitation such as general relativity. It is interesting to note that Christoffel symbols are related to Cartan connection by the following mathematical identity

$$\Gamma_{\mu\lambda\nu} = {}^{0}\Gamma_{\mu\lambda\nu} + K_{\mu\lambda\nu}, \tag{3}$$

where $K_{\mu\lambda\nu}$ is given by

$$K_{\mu\lambda\nu} = \frac{1}{2}(T_{\lambda\mu\nu} + T_{\nu\lambda\mu} + T_{\mu\lambda\nu}), \tag{4}$$

with $T_{\mu\lambda\nu} = e_{a\mu}T^{a}{}_{\lambda\nu}$, the quantity $K_{\mu\lambda\nu}$ is the contortion tensor.

The curvature tensor obtained from $\Gamma_{\mu\lambda\nu}$ is identically zero which, using (3), leads to

$$eR(e) \equiv -e(\frac{1}{4}T^{abc}T_{abc} + \frac{1}{2}T^{abc}T_{bac} - T^{a}T_{a}) + 2\partial_{\mu}(eT^{\mu}), \tag{5}$$

where $R(e)$ is the scalar curvature of a Riemannian manifold and $T^{\mu} = T^{b}{}_{b}{}^{\mu}$. Since the divergence term in Equation (5) does not contribute with the field equations, hence the Teleparallel Lagrangian density equivalent to Hilbert-Einstein Lagrangian density is

$$\begin{aligned} \mathfrak{L}(e_{a\mu}) &= -\kappa e\left(\frac{1}{4}T^{abc}T_{abc} + \frac{1}{2}T^{abc}T_{bac} - T^{a}T_{a}\right) - \mathfrak{L}_{M} \\ &\equiv -\kappa e\Sigma^{abc}T_{abc} - \mathfrak{L}_{M}, \end{aligned} \tag{6}$$

where $\kappa = 1/(16\pi)$, $\mathfrak{L}_{M}$ is the Lagrangian density of matter fields and $\Sigma^{abc}$ is given by

$$\Sigma^{abc} = \frac{1}{4}(T^{abc} + T^{bac} - T^{cab}) + \frac{1}{2}(\eta^{ac}T^{b} - \eta^{ab}T^{c}), \tag{7}$$

with $T^{a} = e^{a}{}_{\mu}T^{\mu}$. The field equations obtained from such a Lagrangian read

$$\partial_{\nu}\left(e\Sigma^{a\lambda\nu}\right) = \frac{1}{4\kappa}e\,e^{a}{}_{\mu}(t^{\lambda\mu} + T^{\lambda\mu}), \tag{8}$$

where $T^{\lambda\mu}$ is the energy-momentum of matter fields while $t^{\lambda\mu}$ which is defined by

$$t^{\lambda\mu} = \kappa\left[4\Sigma^{bc\lambda}T_{bc}{}^{\mu} - g^{\lambda\mu}\Sigma^{abc}T_{abc}\right], \tag{9}$$

represents the gravitational energy-momentum [16]. It should be noted that $\Sigma^{a\lambda\nu}$ is skew-symmetric in the last two indices, that leads to

$$\partial_{\lambda}\partial_{\nu}\left(e\Sigma^{a\lambda\nu}\right) \equiv 0. \tag{10}$$

Therefore the total energy-momentum contained in a three-dimensional volume $V$ of space is

$$P^{a} = \int_{V} d^{3}x\,e\,e^{a}{}_{\mu}(t^{0\mu} + T^{0\mu}), \tag{11}$$

or using the field equations we have

$$P^{a} = 4k\int_{V} d^{3}x\,\partial_{\nu}\left(e\,\Sigma^{a0\nu}\right). \tag{12}$$

It is worth mentioning that the above expression is independent of coordinate transformations, which is expected from a reliable definition of energy and momentum. On the other hand, it is a vector under Lorentz transformations, which is a feature of special relativity and there is no good reason to abandon such an attribute in gravitational theory.

We stress the fact that the tetrad field is the dynamical variable of teleparallel gravity, then the usual definition of angular momentum in terms of the energy-momentum vector yields

$$L^{ab} = 4k \int_V d^3x\, e \left( \Sigma^{a0b} - \Sigma^{b0a} \right),$$

(13)

this expression is the total angular momentum. Both $L^{ab}$ and $P^a$ obey a Poincaré algebra [17], which is a very good indication of the consistency of the definition (13).

## 3. Angular Momentum Quantization

The most general form of the line element that exhibits axial symmetry is given by

$$ds^2 = g_{00}dt^2 + 2g_{03}d\phi\, dt + g_{11}dr^2 + g_{22}d\theta^2 + g_{33}d\phi^2.$$

(14)

That yields the following contravariant metric tensor

$$g^{\mu\nu} = \begin{pmatrix} -\dfrac{g_{33}}{\delta} & 0 & 0 & \dfrac{g_{03}}{\delta} \\ 0 & \dfrac{1}{g_{11}} & 0 & 0 \\ 0 & 0 & \dfrac{1}{g_{22}} & 0 \\ \dfrac{g_{03}}{\delta} & 0 & 0 & -\dfrac{g_{00}}{\delta} \end{pmatrix}$$

(15)

where $\delta = g_{03}g_{03} - g_{00}g_{33}$. it should be noted that the components of the metric tensor are function of $r$ e $\theta$.

In order to calculate the angular momentum we have to choose a referencial frame. As stated before, the kinematical state of the observer can be defined by its field velocity. For a stationary observer it is enough to chose $e_{(0)}{}^i = U^i = 0$, which is also known as Schwinger gauge. In fact, the most general way of establishing the reference frame is through the acceleration tensor first introduced by Mashhoon [18,19]. Thus, a tetrad field adapted to a stationary reference frame is given by

$$e_{a\mu} = \begin{pmatrix} -A & 0 & 0 & -B \\ 0 & \sqrt{g_{11}}\sin\theta\cos\phi & \sqrt{g_{22}}\cos\theta\cos\phi & -C\sin\theta\sin\phi \\ 0 & \sqrt{g_{11}}\sin\theta\sin\phi & \sqrt{g_{22}}\cos\theta\sin\phi & C\sin\theta\cos\phi \\ 0 & \sqrt{g_{11}}\cos\theta & -\sqrt{g_{22}}\sin\theta & 0 \end{pmatrix},$$

(16)

with

$$\begin{aligned} A &= \sqrt{(-g_{00})}, \\ AB &= -g_{03}, \\ C\sin\theta &= \dfrac{\delta^{1/2}}{\sqrt{(-g_{00})}} \end{aligned}$$

(17)

Then the non-vanishing components of the torsion tensor are

$$
\begin{aligned}
T_{013} &= -A\partial_1 B\,, \\
T_{023} &= -A\partial_2 B\,, \\
T_{001} &= \frac{1}{2}\partial_1(A^2)\,, \\
T_{002} &= \frac{1}{2}\partial_2(A^2)\,, \\
T_{112} &= -\frac{1}{2}\partial_2(g_{11})\,, \\
T_{212} &= \frac{1}{2}\partial_1(g_{22}) - \sqrt{g_{11}g_{22}}\,, \\
T_{313} &= \frac{1}{2}\partial_1(g_{33}) - \sqrt{g_{11}}C\sin^2\theta\,, \\
T_{323} &= \frac{1}{2}\partial_2(g_{33}) - \sqrt{g_{22}}C\sin\theta\cos\theta\,.
\end{aligned}
\tag{18}
$$

From expression (13) it is possible to define the angular momentum density $M^{ab} = 4ke\left(\Sigma^{a0b} - \Sigma^{b0a}\right)$, then after some algebraic manipulations it yields

$$
M^{ab} = 2k\partial_i[e(e^{ai}e^{b0} - e^{bi}e^{a0})]\,,
\tag{19}
$$

which is an incredible simple expression in terms of the tetrad field adapted to a stationary reference frame.

The most historical representative of axial symmetry is the Kerr solution. Perhaps it is a natural step to investigate the quantization of angular momentum in such a system. This line element, in terms of the Boyer-Lindquist coordinates, is given by

$$
ds^2 = -\frac{\psi^2}{\rho^2}dt^2 - \frac{2\chi\sin^2\theta}{\rho^2}d\phi\,dt + \frac{\rho^2}{\Delta}dr^2 + \rho^2 d\theta^2 + \frac{\Sigma^2\sin^2\theta}{\rho^2}d\phi^2\,,
\tag{20}
$$

with

$$
\begin{aligned}
\Delta &= r^2 + a^2 - 2mr\,, \\
\rho^2 &= r^2 + a^2\cos^2\theta\,, \\
\Sigma^2 &= (r^2 + a^2)^2 - \Delta a^2\sin^2\theta\,, \\
\psi^2 &= \Delta - a^2\sin^2\theta\,, \\
\chi &= 2amr\,.
\end{aligned}
\tag{21}
$$

This black hole has a fundamental singularity in the form of a ring in contrast with Schwarzschild black hole whose singularity is a point. The Kerr black hole rotates with angular velocity $\omega = \dfrac{2mar\sin^2\theta}{\Sigma^2}$ and has two event horizon given by $r_\pm = m \pm \sqrt{m^2 - a^2}$. In addition, this black hole has two stationary surfaces defined by $r_{s\pm} = m \pm \sqrt{m^2 - a^2\cos^2\theta}$. The region between the event horizon and the stationary surface is called ergosphere, there it is impossible to have a reference frame at rest. In order to simplify our analysis, let us take a slowly rotating Kerr space-time which is given by the following line element

$$
ds^2 = -\left(1 - \frac{2m}{r}\right)dt^2 + \left(1 - \frac{2m}{r}\right)^{-1}dr^2 + r^2 d\theta^2 + r^2\sin^2\theta d\phi^2 - \frac{2ma}{r}\sin^2\theta dt d\phi\,,
\tag{22}
$$

which yields a tetrad field in agreement with reference [20], then the component of angular momentum density in z-direction, $M_z = M_{(1)(2)}$, reads

$$M_z = \frac{ma \sin \theta}{r^2} \left(1 - \frac{2m}{r}\right)^{-3/2} \left[-m \sin^2 \theta + r(3 \cos^2 \theta - 1) \sqrt{1 - \frac{2m}{r}}\right]. \tag{23}$$

Similarly, the modulus of the angular momentum density, $M^2 = M^{ab} M_{ab}$, is given by

$$M^2 = \frac{m^2 a^2 \sin^2 \theta}{r(8m^3 - r^3 + 6mr^2 - 12m^2 r)} \left[r(2m - r)(4 - 3 \sin^2 \theta) - m(2r + m)\left(1 - \frac{2m}{r}\right)^{1/2} \sin^2 \theta\right]. \tag{24}$$

Now we have to apply some quantization procedure to those expressions of angular momentum. Thus, the Weyl quantization is a mapping that leads classical coordinates, $z_n$, into operators $\widehat{z_n}$. Such a map is explicitly given by

$$\mathcal{W}[f(z_1, z_2, ..., z_n)] := \frac{1}{(2\pi)^n} \int d^n k d^n z f(z_1, z_2, ..., z_n) \exp\left(i \sum_{l=1}^{n} k_l(\widehat{z_l} - z_l)\right), \tag{25}$$

usually the operators $\widehat{z_n}$ obey a non-commutative relation as

$$[\widehat{z_i}, \widehat{z_j}] = \beta_{ij}.$$

It is worth mentioning that Weyl's quantization method is suitable to quantize any particular function of coordinates, which is an huge advantage over the canonical procedure that can be applied only in the phase space. This sort of quantization arose in the very dawn of quantum mechanics when the opposite question about the classical limit of a quantum structure was asked. Another interesting point about Weyl's prescription is the non-commutative relation between the operators $\widehat{z_i}$, it allows any representation for such operator as long as the non-commutative relation holds. Thus, it is a matter of convenience on how representation should be used. For instance, in the phase space of quantum mechanics one can use the momentum representation settled by $\mathcal{W}[x] = \widehat{x} = i\hbar \dfrac{\partial}{\partial p}$ and $\mathcal{W}[p] = \widehat{p} = p$, the coordinate representation, which is given by $\mathcal{W}[p] = \widehat{p} = -i\hbar \dfrac{\partial}{\partial x}$ and $\mathcal{W}[x] = \widehat{x} = x$, or a mixture of both. It should be noted that both $M_z$ and $M^2$ depend on the coordinates $r$ and $\sin \theta$, hence we introduce the simpler representation $\mathcal{W}[r] = \widehat{r} = \beta \dfrac{\partial}{\partial x}$ and $\mathcal{W}[\sin \theta] = \widehat{x} = x$, with $\beta_{12} = \beta$. Therefore, applying the Weyl prescription to $\mathcal{W}[M_z] = \widehat{M_z}$ and requiring

$$\widehat{M_z} \psi = \lambda \psi,$$

it yields

$$
\begin{aligned}
\frac{\lambda}{ma}\left(\beta^2 \frac{d^2}{dx^2} - 3m\beta \frac{d}{dx}\right)\psi &= -mx^3 \psi + \frac{1}{2}\left(2x - 3x^3\right)\left(\beta \frac{d}{dx} + 2m\right)\psi \\
&+ \frac{1}{2}\left(\beta \frac{d}{dx} + 2m\right)\left(2x - 3x^3\right)\psi.
\end{aligned}
\tag{26}
$$

Then

$$j\frac{d^2 \psi}{dx^2} + (3x^3 - 2x - 3\mu j)\frac{d\psi}{dx} + \left(\frac{9}{2}x^2 - 2\mu x^3 + 2\mu x - 1\right)\psi = 0, \tag{27}$$

where $\mu = \dfrac{m}{\beta}$ and $j = \dfrac{\lambda}{\mu a}$. It should be noted that $\beta$ is the non-commutative parameter with dimension of length hence $\mu$ and $j$ are dimensionless. We also point out that a quantization process is essentially

the introduction of a non-commutative structure. The Weyl's prescription allows one to construct such a structure including, but not restricted to, the phase space as in quantum mechanics. In addition, the quantization process requires a function of coordinates in which a non-commutative parameter can be introduced, that excludes $L^{ab}$ since it is independent on the coordinates. Although the angular momentum density has no physical meaning under the classical viewpoint, it yields an observable in the realm of quantum theory. For instance $\lambda = \int d^3x \psi^\dagger \hat{M}_z \psi$ which could be experimentally verified.

Analogously, the Weyl procedure applied to $M^2$ together with an equation of eigenfunction and eigenvalue as $\widehat{M^2}\Psi = \alpha^2 \Psi$ yields

$$
\begin{aligned}
l^2 \frac{\partial^4 \Psi}{\partial x^4} &- 6\mu l^2 \frac{\partial^3 \Psi}{\partial x^3} + (3x^4 - 4x^2 + 12\mu^2 l^2) \frac{\partial^2 \Psi}{\partial x^2} + (8\mu x^2 - 8l^2 \mu^3 - 8\mu x^4 - 8x + 12x^3) \frac{\partial \Psi}{\partial x} \\
&+ (\mu^2 x^4 - 16\mu x^3 + 8\mu x + 18x - 4)\Psi = 0,
\end{aligned}
\tag{28}
$$

where $\mu = \dfrac{m}{\beta}$ and $l = \dfrac{\alpha}{\mu a}$. It is worth pointing out that the densities associated to $P^a$ and $L^{ab}$ were obtained as constraints in the Hamiltonian formalism of teleparallel gravity. They satisfy the Poincaré algebra as well, thus $M^{ab}M_{ab}$ is the most suitable SO(3,1) invariant to apply the Weyl prescription since it is still dependent on the coordinates. In order to present a solution for Equations (27) and (28), we will use the Adomian Decomposition Method.

*Adomian Decomposition Method (AM)*

The Adomian Decomposition Method (AM) was developed in 1961 to solve frontier physical problems [21]. The method shows excellent results in the study of nonlinear ordinary differential, integro-differential, and partial differential equations. It is based on the following steps: Consider an equation

$$
Hy(x) = f(x),
\tag{29}
$$

where $H$ stands for a general nonlinear ordinary differential operator, with linear and nonlinear terms. The linear part can be separated in two others, $L$ and $R$, where $L$ is easily inverted and $R$ is the remainder of the linear operator. In this sense, operator $H$ can be written as

$$
H = L + R + N,
$$

where $N$ is the nonlinear part. Then, Equation (29) becomes

$$
Ly(x) + Ry(x) + Ny(x) = f(x).
\tag{30}
$$

Equation (30) can be written as

$$
L^{-1}[Ly(x)] = -L^{-1}[Ry(x)] - L^{-1}[Ny(x)] + L^{-1}[f(x)],
\tag{31}
$$

where $L^{-1}$ is the inverse operator of $L$. If $L$ is a second order operator, for example, we have $L^{-1}[Ly(x)] = y(x) - y(0) - y'(0)x$, and the solution of Equation (31) turns out to be

$$
y(x) = y(0) + y'(0)x - L^{-1}[Ry(x)] - L^{-1}[Ny(x)] + L^{-1}[f(x)].
\tag{32}
$$

The nonlinear term $Ny(x)$ can be expanded as $\sum_{n=0}^{\infty} A_n$, where $A_n$ are the so called Adomian polynomials.

The remaining part $y(x)$ will be decomposed into $y(x) = \sum_{n=0}^{\infty} y_n$, with $y_0 = y(0) + y'(0)x + L^{-1}[f(x)]$. Consequently, we have

$$y(x) = y_0 - L^{-1}[R(\sum_{n=0}^{\infty} y_n)] - L^{-1}[\sum_{n=0}^{\infty} A_n]. \tag{33}$$

The Adomian polynomials can be calculated using the relation

$$A_n = \frac{1}{n}\left[\frac{d}{d\lambda^n}[N(\sum_{n=0}^{\infty} \lambda^i y_i)]\right]_{\lambda=0}, \tag{34}$$

$n = 0, 1, 2, 3, \dots$. An extensive use of such a method can be found in reference [21].

If we apply this method to Equation (27), then we obtain

$$\psi(x) = \psi(0) + \frac{d\psi(0)}{dx}x - \frac{1}{j}\left\{L^{-1}\left[(3x^3 - 2x - 3\mu j)\frac{d\psi(x)}{dx}\right] - L^{-1}\left[(-2\mu x^3 + \frac{9}{2}x^2 + 2\mu x - 1)\psi(x)\right]\right\}. \tag{35}$$

Hence using $\psi(x) = \sum_{n=0}^{\infty} \psi_n$, we have

$$\psi_0(x) = \psi(0) + \frac{d\psi(0)}{dx}x.$$

It should be pointed out that the boundary condition used has a clear meaning, it says that the information about angular momentum is the same at the poles of Kerr ergosphere. The superior orders of approximation in the solution can be found by an iterative procedure, it is given by

$$\psi_1(x) = -\frac{1}{j}\left\{L^{-1}\left[(3x^3 - 2x - 3\mu j)\frac{d\psi_0(x)}{dx}\right] - L^{-1}\left[(-2\mu x^3 + \frac{9}{2}x^2 + 2\mu x - 1)\psi_0(x)\right]\right\},$$

$$\psi_2(x) = -\frac{1}{j}\left\{L^{-1}\left[(3x^3 - 2x - 3\mu j)\frac{d\psi_1(x)}{dx}\right] - L^{-1}\left[(-2\mu x^3 + \frac{9}{2}x^2 + 2\mu x - 1)\psi_1(x)\right]\right\},$$

$$\psi_{n+1}(x) = -\frac{1}{j}\left\{L^{-1}\left[(3x^3 - 2x - 3\mu j)\frac{d\psi_n(x)}{dx}\right] - L^{-1}\left[(-2\mu x^3 + \frac{9}{2}x^2 + 2\mu x - 1)\psi_n(x)\right]\right\}.$$

In this way, the solution of Equation (27) up to the second order approximation is

$$
\begin{aligned}
\psi(x) ={}& \psi(0) + \frac{d\psi(0)}{dx}x + \frac{1}{2}\left(\frac{3\mu}{j}\frac{d\psi(0)}{dx} - \frac{1}{j}\psi(0)\right)x^2 \\
&+ \frac{1}{3}\left(-\frac{3}{2j^2}\frac{d\psi(0)}{dx} - \frac{2\mu}{j^2}\psi(0)\right)x^3 + \frac{1}{4}\left(\frac{1}{j^2}\frac{d\psi(0)}{dx} - \frac{4\mu}{3j^2}\frac{d\psi(0)}{dx} + \frac{1}{j^2}\psi(0)\right)x^4 \\
&+ \frac{1}{5}\left(\frac{3}{4j^2}\frac{d\psi(0)}{dx} + \frac{7\mu}{4j^2}\psi(0)\right)x^5 + \frac{7\mu}{30j^3}\frac{d\psi(0)}{dx}x^6 + \dots
\end{aligned} \tag{36}
$$

With the condition $\psi(1) = \psi(-1)$, we estimate that the lower value for eigenvalue $j$ is $j_1 = \sqrt{\dfrac{3 + 4\mu}{6}}$.
In the second level, the value for eigenvalue $j$ becomes $j_2 = \sqrt{\dfrac{1 + 2\mu}{8}}$. It should be noted that $j$ assumes discrete values depending on the approximation order taken in AM.

Similarly, the solution obtained when AM is applied to (28), again up to the second order approximation, reads

$$
\begin{aligned}
\Psi(x) \;=\; & \Psi(0) + \frac{d\Psi(0)}{dx}x + \frac{1}{2}\frac{d^2\Psi(0)}{dx^2}x^2 + \frac{1}{6}\frac{d^3\Psi(0)}{dx^3}x^3 \\
& + \frac{1}{4}\left(\frac{4}{3}\frac{d\Psi(0)}{dx}\mu^3 - 2\mu^2\frac{d^3\Psi(0)}{dx^3} + \frac{2}{3}\Psi(0)\right)x^4 \\
& + \frac{1}{5}\left(\frac{1}{3}\frac{d^3\Psi(0)}{dx^3}\mu^3 + \frac{1}{5l^2} - \frac{2\mu)}{5l^2}\frac{d\Psi(0)}{dx} + \frac{1}{30}\frac{d^2\Psi(0)}{dx^2} - \frac{1}{60}(8\mu + 18)\Psi(0)\right)x^5 \\
& + \frac{1}{6}\left(\frac{\mu^3}{15}\frac{d^3\Psi(0)}{dx^3} - \frac{1}{5l^2}\frac{d^2\Psi(0)}{dx^2} - \frac{2\mu}{5l^2}\frac{d\Psi(0)}{dx} + \frac{1}{30}\frac{d^2\Psi(0)}{dx^2} - \frac{(18 + 8\mu)}{6}\frac{d^3\Psi(0)}{dx^3}\right)x^6 \\
& + \frac{1}{7}\left(\frac{1}{180}\frac{d^3\Psi(0)}{dx^3} + \frac{1}{15l^2}\frac{d^3\Psi(0)}{dx^3} - \frac{1}{10l^2}\frac{d\Psi(0)}{dx} + \frac{2\mu}{15}\Psi(0) - \frac{(18 + 8\mu)}{240}\frac{d^2\Psi(0)}{dx^2} - \frac{\mu}{15l^2}\frac{d^2\Psi(0)}{dx^2}\right)x^7 + ...
\end{aligned}
\tag{37}
$$

This solution gives a discrete value of angular momentum as well as above. The lowest level of eigenvalue $l^2$ is given by $l_1^2 = \dfrac{8\mu^3 + 12\mu^2 - 8\mu - 14}{24}$. It is worth pointing out that $l_n$ and $j_n$ stand for discrete dimensionless values for the total gravitational angular momentum and gravitational angular momentum in z-direction respectively. It means that if the black hole mass can continuously grow then the gravitational angular momentum assumes only allowed values.

## 4. Conclusions

In this article we have presented the quantization of angular momentum in a slowly rotating Kerr space-time. The angular momentum was obtained in the realm of teleparallel gravity which separates features of gravitational field from matter fields. A slowly rotating approximation was used to calculate the angular momentum and the Weyl quantization procedure was used to obtain a quantum equation. Such a prescription was applied to the z-direction of gravitational angular momentum and its squared. The equations were established by an eigenvalue-eigenfunction equation. We used the Adomian method to obtain an approximated solution and, using boundary conditions, we find out a discrete angular momentum eigenvalue. Such a discrete feature can help us to look for experimental evidences of a quantum theory of gravitation.

**Author Contributions:** Conceptualization, S.U. and E.S.; methodology, R.A.; software, R.A.; validation, S.U., E.S. and R.A.; formal analysis, E.S. and R.A.; investigation, S.U., E.S. and R.A.; resources, S.U. and E.S.; data curation, E.S.; writing—original draft preparation, S.U.; writing—review and editing, S.U.; visualization, S.U., E.S. and R.A.; supervision, S.U.; project administration, not applicable; funding acquisition, not applicable.

**Funding:** This research received no external funding.

**Conflicts of Interest:** The authors declare no conflict of interest.

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
