# Peer review of "On Quantization of a Slowly Rotating Kerr Black Hole in Teleparallel Gravity"

_universe, doi:10.3390/universe5010029_

Round 1
Reviewer 1 Report
I have read the manuscript “On Quantization of a Slow Rotating Kerr Black Hole” by Ulhoa et al.. The authors studied the the quantization of slow rotating in Kerr metric in the context of teleparallel gravity, and using the Weyl formalism. This paper is an incremental work to the authors’ previous work published in Ref [11] Ulhoa and Amorim. Adv. High Energy Phys., 2014:1, 2014. There are a number of issues in the paper.
1- The abstract is not promising. They did not properly mention the method, i.e. “teleparallel gravity”.
2- Introduction. The author did not give enough background to teleparallel gravity and Kerr metric in teleparallel gravity. There are many review papers, which the authors could cite them, such as
de Andrade, Guillen, Pereira, arXiv:gr-qc/0011087
Cai, Capozziello, De Laurentis, Saridakis, arXiv:1511.07586
Arcos, Pereira, arXiv:gr-qc/0501017
Garecki, arXiv:1010.2654
Aldrovandi and Pereira, Teleparallel Gravity: An Introduction, Springer 2012.
For a Kerr metric in teleparallel gravity, for example,
Pereira, Vargas, Zhang, arXiv:gr-qc/0102070
3- The authors could include a subsection in Section 2 which gives some background on Weyl formalism. Currently, some of them can be found in Section 3, e.g. eq (23).
4- Section 3, the authors need to discuss how their results are related to the previous results of Kerr metric in teleparallel gravity, e.g. Pereira, Vargas, Zhang, arXiv:gr-qc/0102070
5- There are a few phrase correction, e.g.
"riemannian geometry" -> "Riemannian geometry"
“weitzenböckian manifold” -> “Weitzenböck manifold”
“Adomian method” -> “Adomian decomposition method” or “ADM”
Author Response
Please see attached responses to reviewers.

Reviewer 2 Report
Angular-momentum quantization of the Kerr solution in Teleparallel
gravity is analyzed.
Teleparallel gravity is revised, and, in particular, the
methods for the quantization of angular momentum.
Major Revisions
1 the acronym ADM in paragraph 3.1 already
stands for Arnowitt-Deser-Misner
(decomposition of the metric tensor in General Relativity)
and should be replaced by
a different
acronym in order to avoid confusion;
2 the expression 'slow rotating' should be replaced by 'slowly-rotating'
throughout the paper;
3 another compilative Section reviewing Kerr blackholes should be added;
4 the expression 'in Teleparallel gravity' should be added
at the end of the title of the paper to better clarify the
content of the paper

Author Response

(The authors gave the same response as above.)

Reviewer 3 Report
This article discusses angular momentum of rotating black holes within the formalism of teleparallel gravity. The presentation on pages 1-4 is clear, but the transition from equation (19) to (22) is not justified, and this casts doubt on correctness of all statements made between equation (22) and the end of the paper. Specifically, starting with a reasonable expression (13) for the total angular momentum, the authors define the density (19). One can also define the square of the total angular momentum as L_{ab}L^{ab}, but its density would not be given by equation (22) (density of a square is not the same as square of density). Thus the rest of the paper dealing with properties of (22) appears to be irrelevant for the study of angular momentum, so all arguments presented on pages 5-7 should be rewritten. In addition to this main mistake, there are some other problem with the paper:
1. In section 2 the authors build angular momentum from a stress-energy tensor of gravitational field. It is well known that gravity admits only stress-energy pseudo-tensor, and only the integrals, such as (12) and (13) have tensor properties. The density (19) is not guaranteed to have tensor transformation laws, and the authors should say something about this.
2. As indicated above, in gravitational theories densities do not transform well, so usually only asymptotic charges are defined. This can be done either through (12) and (13) or through Arnowitt-Deser-Misner (ADM) formalism. Contrary to the existing literature, the authors seem to imply that densities of conserved charges have physical meaning, and much more elaboration on this statement is needed. Incidentally Arnowitt-Deser-Misner formalism is not mentioned at all, and ADM abbreviation is used for something else.
3. Equation (22) is the main problem with this article, this may be a density of something, but clearly not one of L_{ab}L^{ab}. Justification of (22) is needed.
4. Even assuming that the spectrum of (22) is interesting, the differential equations written for that operator require boundary conditions, including one at the horizon. As phenomenon of Hawking radiation indicates, such conditions must be imposed with great care, and authors certainly did not do that. Even after replacing (22) by its correct version, the authors should change the logic in deriving equations (24)-(34) to account for the boundary conditions.
To summarize, this paper contains misleading statements and wrong calculations, so I do not recommend it for publication.
Reviewer 4 Report
The manuscript in question is very interesting and in a sense deals with an important question of the quantum nature of gravity.
Even though the manuscript is interesting and presents some new ideas I can not recommend its publication until a certain revision is made. Here are some of my comments:
1) In the last two sentences of the Abstract the english is not so good. Expressions like "...by means the ..." and "...we find out a..." are strange to say the least.
2)Introduction, second line : of a -> from
4th line: expression -> quantization
3) The text from line 9 to 19 needs an improvement in english.
4) sentence in line 20 is strangely formulated.
5) line 46: Weitzenbockian, 48: Riemannian
6) section 2 should be written with more details. Namely the starting point of the Teleparallel Gravity should be better explained.
7) In line 78 the stationary reference frame is introduced without any explanation, definition, properties or motivation. Few comments are needed.
8) The part from eq.(21) to eq.(25) is very unclear. First of all, a step-by-step derivation of at least equation (24) is needed. Secondly, the representation in line 95 is unclear. Namely, here the classical coordinates r and sin\theta are promoted into operators such that we have
[W(r),W(sin\theta)]=\beta
...this is basically quantizing space itself, that is more in the sense of Noncommutative geometry....one is not quantizing phase space. What is the motivation for such a representation? And what is the physics of the parameter \beta....which now seems to be of the dimension of length.
So, an explanation of why is one using [W(r),W(sin\theta)]=\beta and a detail derivation of at least (24) is needed.
Also, sense one is dealing with quantized or better to say noncommutative space, the authors should refer to some of the references on that topic (especially on the ones dealing with noncommutative spaces and black holes) .
9) The final result of the paper seams to be reasonable (modulo the comments aforementioned) but the author should stress more on the physical meaning of the discrete spectrum of the angular moment. Could one obtain the quantization of mass/energy if one uses the eq(11) and apply the methods using eq. (23).
Round 2
Reviewer 1 Report
The authors addressed my comments, now it is suitable for publication.
Author Response
Thanks.
Reviewer 2 Report
The Authors have accomplished with the suggestion tasks
Author Response
Thanks.
Reviewer 3 Report
I thank the authors for making minor changes to the manuscript, but the criticism raised in my original report has not been addressed. Specifically, my main point
"One can also define the square of the total angular momentum as L_{ab}L^{ab}, but its density would not be given by equation (22) (density of a square is not the same as square of density). Thus the rest of the paper dealing with properties of (22) appears to be irrelevant for the study of angular momentum, so all arguments presented on pages
5-7 should be rewritten."
implies that the investigation undertaken in this article is irrelevant for the study of angular momentum, and author's response "It is impossible to obtain a quantum equation from L_{ab} since it does not depend on the coordinates, which means there is no possible operator to be introduced." does not change this fact. The old equation (22) has been removed (or hidden from the reader), but the wrong conclusions stemming from it are still claimed to be the main result of this article.
Other points have not been properly addressed as well, for example, the statement from the reply "We point out that the representation of non-commutative operators used makes the suggested boundary condition at the horizon impracticable." suggests that the Hawking radiation, the most interesting effects associated with black holes, would be "impracticable" in the formalism used by the authors. This article is misleading, and I do not recommend it for publication.
Author Response
We find it impossible to address the referee’s suggestions, please attached is the letter with our arguments. In our opinion the analysis of Hawking radiation requires the introduction of temperature which is not the scope of our article. In addition we don’t think that the quantum invariant should be defined from $L^{ab}L_{ab}$ firstly because there no clear density associated to it and secondly it is not useful in the quantization method used in the article.

Reviewer 4 Report
The author's response is satisfactory and therefore can recommend the paper for publication.
Author Response
Thanks.